# Ammonium Catecholaldehydes as Multifunctional Bioactive Agents: Evaluating Antimicrobial, Antioxidant, and Antiplatelet Activity

**DOI:** 10.3390/ijms26167866

**Published:** 2025-08-14

**Authors:** Andrei V. Bogdanov, Roza G. Tagasheva, Alexandra Voloshina, Anna Lyubina, Olga Tsivileva, Artem N. Kuzovlev, Wang Yi, Aleksandr V. Samorodov, Guzel K. Ziyatdinova, Elnara R. Zhiganshina, Maxim V. Arsenyev, Sergey V. Bukharov

**Affiliations:** 1Department of High Molecular and Organoelement Compounds, Analytical Chemistry Department, Kazan Federal University, Kremlevskaya Str. 18, Kazan 420008, Russia; guzel.ziyatdinova@kpfu.ru; 2Department of Technology of Basic Organic and Petrochemical Synthesis, Kazan National Research Technological University, K. Marx Str. 68, Kazan 420015, Russia; roza-ta1982@yandex.ru (R.G.T.); svbukharov@mail.ru (S.V.B.); 3Arbuzov Institute of Organic and Physical Chemistry, FRC Kazan Scientific Center, Russian Academy of Sciences, Akad. Arbuzov St. 8, Kazan 420088, Russia; microbi@iopc.ru (A.V.); aplyubina@gmail.com (A.L.); 4Institute of Biochemistry and Physiology of Plants and Microorganisms, Saratov Scientific Centre of the Russian Academy of Sciences, Entuziastov Ave. 13, Saratov 410049, Russia; tsivileva_o@ibppm.ru; 5V. A. Negovsky Research Institute of General Reanimatology, Russian Academy of Medical Sciences, Petrovka St. 25/2, Moscow 107031, Russia; artem_kuzovlev@fnkcrr.ru; 6School of Pharmacy, Hangzhou Normal University, Yuhangtan, 2318, Hangzhou 310030, China; yi.wang1122@hznu.edu.cn; 7Department of Pharmacology, Bashkir State Medical University, Lenin St. 3, Ufa 450008, Russia; 8G. A. Razuvaev Institute of Organometallic Chemistry, RAS, Tropinin Str. 49, Nizhny Novgorod 603950, Russia; zhiganshinae@mail.ru (E.R.Z.); mars@iomc.ras.ru (M.V.A.)

**Keywords:** hydrazones, ammonium salts, catechols, antioxidants, phytopathogens, antimicrobial activity

## Abstract

A wide range of water-soluble quaternary ammonium acylhydrazones based on catecholaldehyde were synthesized and characterized using NMR, IR spectroscopy, and elemental analysis. The total antioxidant capacity of the acylhydrazones discussed herein was estimated via coulometric titration with electrogenerated bromine. Pyridinium derivatives **11a**–**e** exhibited the highest antioxidant capacity. Quaternary ammonium acylhydrazones demonstrated high antimicrobial activity against Gram-positive bacteria, including methicillin-resistant *Staphylococcus aureus* strains. Furthermore, low hemo- and cytotoxicity and the absence of a negative effect on the hemostatic system were confirmed for the studied compounds. According to the results of a CV test, the antimicrobial effect of the most active acylhydrazones, namely, **9a**, **10b**, **10c**, and **11a**, is associated with the destruction of the bacterial cell wall. High or moderate activity against phytopathogens of bacterial origin was observed for all the acylhydrazones evaluated. Anti-aggregation activity was observed for compound **10b**; the extent was 1.6-fold greater than that exhibited by acetylsalicylic acid. On the contrary, compound **9d** exhibited a pro-aggregant effect (with a 6.3% increase in platelet aggregation and a >15% decrease in the latent period compared to the control). Thus, the data obtained can be considered the basis for further pharmaceutical development of these effective drugs with antithrombotic and hemostatic potential.

## 1. Introduction

Bacterial infections of anthropogenic and phytopathogenic origin pose a significant threat to the global population [1,2,3]. The widespread prevalence of drug-resistant pathogens and their high toxicity make currently used drugs ineffective. Therefore, developing novel multi-target agents [4,5,6,7,8] and finding the key structure are crucial tasks.

Natural products have played an essential role in drug design and the treatment of human diseases since ancient times. Amines are among the most ubiquitous functional classes of bioactive natural products and pharmaceuticals. Nevertheless, the enzymatic metabolism of primary amines leads to the formation of endogenous biogenic aldehydes, which can become toxic via covalent modification of proteins, lipids, and DNA [9]. For example, catecholaldehydes and hydrogen peroxide are produced in the monoamine-oxidase-catalyzed oxidative deamination of norepinephrine and dopamine [10]. A nucleophilic hydrazine moiety reacts with free and protein-linked aldehydes, forming stable hydrazones [11]. The beneficial traits of hydralazine in regard to biogenic aldehyde sequestration, particularly with respect to the prevention of oxidative stress development and vascular damage, have been confirmed in animal models [9].

Hydrazones’ versatile biological activities, including their neuroprotective properties, make them promising compounds applicable to the treatment of neurological diseases [12]. *N*-acylhydrazones show potential iron-chelating ability and, therefore, have been suggested as drug candidates for the treatment of iron-overload diseases. *N*-acylhydrazones have a moderate capacity to bind biogenic metals [13]. The affinity and specificity of this process can be controlled by selecting adequate aldehyde precursors and substituents. Studies on pyrocatechol derivatives containing a hydrazone fragment have revealed several prospective compounds with various types of bioactivities. For example, compound **1** (Figure 1) has an excellent neuroprotective effect [14]. Compound **2** demonstrates high antioxidant activity and a potential protective effect against degenerative processes such as age-related macular degeneration [15]. Ammonium hydrazones of pyrocatechol 3 show significant activity against some Gram-positive pathogens [16].

The acylhydrazone platform is a generally recognized structure in medicinal chemistry that has great potential in the creation of new drugs [17,18,19,20]. Molecular hybridization of the acylhydrazone platform with pharmacophore fragments is one of the ways to increase pharmacological efficiency and to find a new generation of agents for combined drug therapy.

Analogues of Girard’s reagents—acetohydrazides containing ammonium fragments—are promising synthetic scaffolds for the creation of new biologically active substances with improved bioavailability [21,22]. The successful use of quaternary ammonium compounds (QACs) in both industry and medicine [23,24] has created opportunities for further development in this field. The possibility that QACs will electrostatically interact with negatively charged surfaces facilitates a wide range of applications [25], including use as anti-corrosion and antistatic agents, employment in air conditioners, and use as active ingredients of disinfectant and biocidal compositions [26,27]. Special attention has been paid to maintaining a balance between QACs’ high activity and their biocompatibility, along with the toxicological aspect, focusing on the effect of QACs on living organisms and the environment [28]. Therefore, the amount of research on QACs is increasing annually, with the main objectives being uncovering the interrelationship between their structure, activity, selectivity, and toxicity toward bacterial and mammalian cells [29,30] and synthesizing a new generation of antimicrobial agents based on QACs [31,32].

As shown in the clinical-trial stage, acylhydrazone derivatives with an *ortho*-hydroxyphenyl group do not have a toxic effect on normal cells and exhibit synergism with numerous cytotoxins in various types of cancer cells [33]. Substituted catechols exhibit various types of physiological activity, including antioxidant properties [34,35,36,37,38,39,40].

The antioxidant and, consequently, biological activity of phenols depends on the steric shielding of their hydroxyl groups. In this work, we investigated the effect of the degree of steric shielding on the biological activity of the most active of the previously studied [16,41] acylhydrazones of hydroxybenzaldehydes and pyrocatechol aldehyde with trialkylammonium and dimethylpyridinium fragments. For this purpose, a set of new ammonium acylhydrazones based on catechol aldehyde substituted with various bulky groups was synthesized, and their antioxidant, antimicrobial, and antiphytopathogenic activities and hemo- and cytotoxicity were studied.

## 2. Results and Discussion

### 2.1. Synthesis

In continuation of our studies on the biological activity of water-soluble phenolic *N*-acylhydrazones [16,41,42], a representative series of new compounds, **9**–**11**, bearing ammonium fragments with different structures, was synthesized. Compounds from series **9** contain a trialkylammonium center, derivatives from series **10** are monocharged bicyclic dibasic compounds, and series **11** constitutes monocyclic pyridinium derivatives (Table 1). Catecholaldhydrazones **9**–**11** were synthesized via an acid-catalyzed condensation reaction of 2,3-dihydroxy-4,6-dialkylbenzaldehydes **1**–**5** with ammonium acetohydrazides **6**–**8** (Figure 1). This approach allowed for an assessment of the contribution of both the structure of the quaternary ammonium cation and the substituents in the catechol core to the corresponding type of biological activity.

Pure forms of ammonium catecholaldhydrazones **9**–**11** were isolated in high yields immediately after reaction media workup. The structure and purity of the compounds were unequivocally verified via IR and NMR spectroscopy and elemental analysis (Appendix A).

For acylhydrazones of carbonyl compounds, spatial structures can be created owing to *syn*/*anti*-isomerism of the amide group (hindered rotation around the N-C(O) bond), inhibited rotation around the N–N bond, and geometric isomerism (*E*/*Z*) at the C=N double bond [43]. *Z*_N–N_ conformation of hydrazones, as a rule, cannot be achieved owing to steric hindrance, leading to disruption of the C=N–NH fragment’s coplanarity and, consequently, n–π conjugation [44]. Thus, the creation of several spatial forms can be expected for a series of new acylhydrazones **9**–**11** (Figure 2).

As shown previously [44], only *E*-isomers of acylhydrazones are formed in the crystalline state. Acylhydrazones mainly form *E*-isomers (up to 100%) in DMSO solutions. The ratio of *anti*- to *syn*-conformers for hydrazones obtained from 2-hydroxybenzaldehydes depends on the strength of the intramolecular hydrogen bond between the nitrogen atom of the imino group and the proton of the 2-hydroxy group [45,46]. At the same time, the signals of the protons from the CH_2_CO and N=CH groups of *anti*-conformers in the ^1^H NMR spectra of acylhydrazones are positioned in the high and weak fields, respectively, compared to the signals of *syn*-conformers [47,48,49,50]. In the ^13^C NMR spectra, the signals of the carbonyl carbon atoms of acylhydrazone *syn*-conformers are down-field-shifted by 5–6 ppm compared to those of *anti*-conformers.

The ^1^H NMR spectra of compounds **9**–**11** show up-field-shifted signals of the methylene protons of the CH_2_CO group of the predominant isomers. The N=CH and OH protons bound by intramolecular hydrogen bonds to the nitrogen atom of the imino group are down-field-shifted compared to the signals of similar protons of minor isomers. The signal of the carbonyl carbon atom in the ^13^C NMR spectra of the major isomer is in a stronger field than the signal in the minor analog. Thus, ammonium acylhydrazones **9**–**11** are a mixture of geometric isomers, with *E*,*anti* as a major isomer and *E*,*syn* as a minor one (Figure 3).

Thus, various water-soluble catecholaldhydrazones were obtained for the first time using easy-to-obtain starting materials and simple and efficient synthetic procedures. These compounds can be divided into three main groups depending on the structures of the substituents at the quaternary nitrogen atom and in the catechol core. The biological activity of water-soluble catecholaldhydrazones was evaluated in the next part of this study.

### 2.2. Biological Studies

The biological activity of the target compounds was evaluated from several perspectives, particularly with respect to their roles as potential antioxidant, antimicrobial, and antiphytopathogenic agents. The novel compounds’ influence on the hemostasis system was also studied (Figure 4).

#### 2.2.1. Antimicrobial Activity

The synthesized series of hydrazones was tested for antimicrobial activity against Gram-positive bacteria: *Staphylococcus aureus* (*Sa*), *Bacillus cereus* (*Bc*), and methicillin-resistant strains of *S. aureus MRSA-1* and *MRSA-2*. The fluoroquinolone antibiotic norfloxacin was used as a reference drug. The results of the analysis of the antimicrobial activity of compounds **9a**–**11e** are presented in Table 2 as the values of the minimum inhibitory concentration (MIC) and minimum bactericidal concentration (MBC). In the tested series, the leading compounds, **9a**, **10b**, and **11a**, were identified, showing antimicrobial activity at the level of the reference drug against *Sa* and *Bc*. The leading compounds also showed high activity against *MRSA-1* and *MRSA-2*. Particularly noteworthy is the DABCO derivative **10b**, which turned out to be the most active against all the bacterial strains, including the *MRSA* strains. It is important to note that compounds **9a**, **10b**, and **11a** exhibited bactericidal action; i.e., their MIC and MBC values differed from each other by no more than four times.

#### 2.2.2. CV Assay

The bacterial cell wall is a target for many antimicrobial agents. Gram-positive bacteria can easily be perceived using the basic dye crystal violet (CV), which is able to bind to the structural elements of the cell wall of viable cells, but when the cell is damaged, the cell wall loses this ability, and the CV is ejected. Therefore, to determine the possible mechanism of the antimicrobial action of the most active ammonium salts, namely, **9a**, **10b**, **10c**, and **11a**, their effect on the cell wall of *Sa* was studied (Figure 5).

Cell wall permeability studies of *Sa* were carried out in the MIC and MBC ranges of the tested compounds. The degree of damage to the bacterial cell wall was assessed according to the percentage of CV in the supernatant. Figure 4 demonstrates the effect of compounds **9a**, **10b**, **10c**, and **11a** on the cell wall of *S. aureus*. Evidently, the most minor changes in the structure of the cell wall occurred when it was exposed to compound **9a**. Thus, at concentrations corresponding to the MIC and MBC (2.2–17.5 µM), the CV yield did not exceed 4%. Significant disturbances were observed only when this compound was added in very high concentrations (C_9a_ ≥ 70 µM). A similar situation occurred after the incubation of *S. aureus* with hydrazones **10b** and **11a**. At concentrations of 2.2–17.5 µM, the percentage of CV did not exceed 20%. However, in contrast to 9a, the CV yield increased significantly even at a concentration of 35 µM, amounting to 86.4% and 86.2%, respectively. The strongest effect was observed after the cells were incubated with compound **10c**. The percentage of CV in the supernatant at a concentration not exceeding the MIC (17.5 µM) was 82.3%. The results obtained indicate that exposure to compounds **9a**, **10b**, and **11a** within the MIC and MIC ranges did not damage the *S. aureus* cell wall. The use of compound **10c** at a concentration ≤ MIC led to significant changes in the structure of the cell wall, which may be due to the structural features of this compound. Finding an answer to this question requires deeper and more detailed research.

#### 2.2.3. Hemolytic and Cytotoxic Activities

When testing new antimicrobial agents, the most important indicator is their toxicity toward mammalian cells in in vitro tests. Therefore, we assessed the hemolytic and cytotoxic activity of the leading compounds.

Experiments were carried out on human red blood cells and human embryonic lung cell culture WI38 (Figure 6) The antibiotic gramicidin S (Gram S), which has a membranotropic effect and moderately high hemolytic activity, was used as a comparison drug to assess hemolysis. When determining the cytotoxic effect of the studied compounds, the well-known cytotoxic anthracycline antibiotic doxorubicin (Dox) was used for comparison. The HC_50_ values of compounds **9a**, **10b**, **10c**, and **11a** were found to be higher than those of gramicidin S (i.e., they exhibited lower hemolytic activity). In relation to human embryonic lung cell culture WI38, all the tested compounds demonstrated low cytotoxic activity compared to doxorubicin. The data obtained indicate the safety of the leading compounds in relation to biological objects and allow us to consider them to be a promising basis for the search for new antimicrobial agents.

#### 2.2.4. Anticoagulant and Antiplatelet Activity Studies

The anticoagulant and antiplatelet properties of the newly synthesized compounds were also studied. The screening data are presented in Table 3.

Compound **10b** exhibited 1.6-fold greater anti-aggregation activity than acetylsalicylic acid (21.8% vs. 13.7%, *p* < 0.05). Among the compounds studied, derivatives **9a**–**c**, **10c**, and **11a**–**c**,**e** exhibited antiplatelet activity matching that of acetylsalicylic acid, reducing maximum platelet aggregation by an average of 10%. However, at the same time, compounds **9a**,**c**,**e**, **10a**–**c**,**e**, and **11a**–**e**, in contrast to acetylsalicylic acid, lengthened the lag period (“platelet release reaction”) relative to the control, indicating the potentially broad antiplatelet potential of the synthesized connections. It should be noted that compound **9d** exhibited a pro-aggregant effect, increasing platelet aggregation by 6.3% (*p* < 0.05) and reducing the latent period by more than 15% (*p* < 0.05) compared to the control. In terms of anticoagulant properties, every compound exhibited different degrees of influence on the plasma component of the hemostasis system, which was manifested in a change only in the indicator of the internal blood coagulation pathway—activated partial thromboplastin time (aPTT). Thus, the studied compounds have high potential as a basis for the development of effective anticoagulants and antiplatelet agents.

#### 2.2.5. Evaluation of Antiphytopathogenic Activity

Spoilage of cultivated plant material, such as fruits, vegetables, and cereals, by phytopathogenic microorganisms routinely occurs all over the world. Infection may be initiated by numerous well-characterized bacterial pathogens, including the genera *Micrococcus*, *Pseudomonas*, *Pectobacterium* (*Erwinia*), and *Xanthomonas* [51]. The top five cereal bacterial pathogens list includes *Pseudomonas* and *Xanthomonas* [52]. *Micrococcus luteus* pathogenic bacterial species are distributed widely in the environment, and, along with *Pectobacterium carotovorum* subsp. *carotovorum*, cause various infections [53]. *Pectobacterium atrosepticum* is known to be a causative agent of soft rot infection in potato tubers [54]. Hence, it is critical to find novel antibacterial agents.

A future direction toward an eco-friendly, sustainable approach to bacterial phytopathogen management may be the implementation of external analogs of the low-molecular-mass substances up-regulated in the infected plants compared to the healthy ones. These substances include *ortho*-diphenols [55], whose structures engage in hydrogen donation, i.e., the ability to improve radical stability by forming an intra-molecular hydrogen bond between the hydrogen of their hydroxyl group and their phenoxyl radicals.

Many synthesized hydrazone derivatives have been explored, particularly in supramolecular and coordination chemistry, materials chemistry, and fields involving complexation with metals [56]. Hydrazones are commonly referred to as potent NH-based antioxidants. Explorations of the diaryl hydrazones have reported that they possess radical-scavenging abilities similar to or greater than those of known reference antioxidants [57]. Importantly, the unsubstituted diaryl hydrazone exhibits only moderate activity, indicating that the substituent effects play a decisive role here [58].

*Ortho*-diphenols and *ortho*-quinones frequently exhibit coupled antioxidant and antibacterial activity [59,60]. Some phenolic-functionalized macromolecules can acquire highly efficient free-radical-scavenging properties. The feasible free-radical-scavenging mechanism includes, among other components, the presence of an *ortho*-dihydroxy substitution, which plays an important role in antioxidant capacity. With this substitution, *ortho*-diphenol is a more efficient moiety than monophenol in terms of the improvement of antioxidant capacity [61]. One *ortho*-substituted phenolic compound, the polyphenol gallic acid, is frequently discussed as perhaps the most important of the phenolic acids owing to its broad spectrum of physiological effects, ranging from antioxidative to antimicrobial in nature [62].

We examined the synthesized sterically hindered 3,5-substituted *ortho*-diphenols and water-soluble hydrazones on the basis for their antibacterial effects on the test organisms noted above. Along with the solutions of **9a**–**e**, **10a**–**e**, and **11a**–**e**, the test compounds sodium hypochlorite (1000 µg/mL), chlorohexidin (500 µg/mL), and standard antibiotics such as ampicillin (**A1**), chloramphenicol (**A2**), kanamycin (**A3**), tetracycline (**A4**), and norfloxacin (**A5**) were used as positive controls at similar dilutions against the bacteria. The in vitro bactericidal activity of the target compounds **9a**–**e**, **10a**–**e**, and **11a**–**e** against five representative phytopathogens is summarized in Table 4.

The results show that compounds **9a**–**e**, **10a**–**e**, and **11a**–**e** exhibited high bactericidal activity against *M. luteus* B-109, *P. atrosepticum* 1043, *P. carotovorum* subsp. *carotovorum* MI, *Ps. fluorescens* EL-2.1, and *X. campestris* B-610 (Table 4). The exceptions were compounds **9e** (inhibition zone width of 3 to 5 mm) and **9c** (inhibition zone width of 7 to 9 mm), which showed a lesser though rather significant bactericidal activity superior to that of the common antibiotic norfloxacin and, in some cases, kanamycin and tetracycline (Table 4). Besides, one worse result was observed in relation to **10e** when applied to *X. campestris*, with the inhibition zone width being 6 mm. Compounds **9a**,**b**,**d**, **10a**–**d**, and **11a**–**e** constitute the group with the most pronounced antibacterial properties, for which the inhibition zone value was not less than 8 mm (Figure 7).

Compounds **9a** and **11a** could be referred to as highly potent bactericidal hydrazones characterized by an inhibition zone width of 14 to 18 mm in the well-agar diffusion assay, and structurally by the presence of a meta-isopropyl moiety adjacent to the *ortho*-diphenol substitution on the aromatic ring.

Superoxide acts as a precursor to other oxidizing agents, such as singlet oxygen and peroxynitrite [63]. Polyphenolic compounds characterized by *ortho*-diphenol moieties may have strong activity in scavenging the superoxide radical anion O_2_^•−^, where quinone–hydroquinone π-conjugated redox systems play an important role. However, when reacting with *ortho*-diphenol, the associated reactions yield highly toxic molecules, such as hydrogen peroxide, which greatly contributes to the *ortho*-diphenols’ antibacterial properties. Thus, the quasi-reversible dioxygen/superoxide redox system is significantly affected by the presence of *ortho*-phenolic groups (catechol-type moiety CatH_2_) engaged in the associated reaction: O_2_^•−^ + CatH_2_ → H_2_O_2_ + Cat^•−^. This reaction is governed by one-step one-electron transfer concerted with two phenolic proton movement after the initial formation of the hydrogen-bonded complexes between superoxide and CatH_2_ [63].

Quinhydrone-type complexes owe their stability to the charge-transfer interactions between hydroquinone (an electron donor) and quinone (an electron acceptor), as well as to hydrogen bonding between the free hydroxyl group of hydroquinone and the carbonyl group of quinine [64]. Recently, a new type of *ortho*-quinhydrone complex was observed for the medicinally important compound desloratadine containing the 4-*tert*-butyl-*ortho*-benzoquinone moiety adjacent to a secondary amine [65]. This synthesized compound interacted non-covalently via its quinone part with 4-*tert*-butylcatechol (initial reactant) through the formation of a hydrogen bond with a free hydroxyl group of the latter and was converted into a new type of *ortho*-quinhydrone complex. In vitro antibacterial studies showed that the resulting *ortho*-benzoquinhydrone complex exhibited a rather good activity (at a level of 30 µg/mL) against the Gram-positive bacteria *Staphylococcus aureus* and *Bacillus cereus*.

*Ortho*-diphenols serve as substrates for microorganisms’ polyphenol oxidases, which are responsible for catalyzing the oxidation of monophenols and/or *ortho*-diphenols into *ortho*-quinones. Through secondary reactions, *ortho*-quinones can promote quinone toxicity and confer resistance against pathogens [66]. Some of the specific activities of polyphenoloxidase are exemplified by isoflavanol dehydratase, which carries out the final ring-closure step of the biosynthesis of the antimicrobial compound, phytoalexin produced in response to phytopathogen infection [67].

Considering the data on the values of the inhibition zone width (mm) (Table 4), the studied equimolar solutions of compounds **9a**–**e**, **10a**–**e**, and **11a**–**e** can be conditionally arranged in descending order of activity:

**9a** (15–18 mm) > **11a** (14–17 mm) > **11b** (9–13 mm) > **9b** (11–12 mm) > **9d** (10–12 mm) > **11e** (8–12 mm) > **10d** (7–12 mm) > **10a** (8–11 mm) > **11c** (7–11 mm) ≈ **10c** (7–11 mm) ≈ **10b** (7–11 mm) > **10e** (6–11 mm) > **11d** (7–9 mm) ≈ **9c** (7–9 mm) > **9e** (3–5 mm).

The hydrophobicity of the compounds tested is thought to be of great importance here. The derivatization of some of the hydrophilic groups improves lipophilicity and allows for better overall lipid permeability [68]. Therefore, it was not surprising that the hydrazone derivatives containing isopropyl, *tert*-pentyl, and methylcyclohexane moieties in their structures appeared to be the most active. Longer chains cause more steric hindrance, while being better suited for lipid permeation [69] to interfere the pathogens survival. The multifaced steric effects could be suggested to impede the mobility of the distinctly substituted hydrazones we studied.

The data we derived from the well-agar diffusion method testify to the poor prospects for the novel antibacterial agent’s development to combat phytopathogens on the basis of adamanthyl (**9e**, **10e**, **11e**) and cyclohexyl (**9c**, **10c**, **11c**) derivatives. Accordingly, these compounds were not involved in the following research. Further, the values of minimal inhibitory concentration (MIC) and minimal bactericidal concentration (MBC) were measured for the compounds in question (Figure 8).

Figure 8 provides a general idea of the MIC and MBC variables measured (at the same time) for all the compounds in the five bacterial test systems in question. This overall picture reveals rather profound species-specific variations in both the bacteriostatic and bactericidal activity observed through the MIC and MBC, respectively, for the standard compounds **A2**, **A4**, and **A5**. Besides, **10a** exhibited poorer performance than the other experimental compounds, most of whose MIC and MBC quantities were below 7 mg/L and 40 mg/L, respectively, therewith being dependent only slightly on the bacterial test system. To estimate the experimental errors omitted in Figure 8 to make the material easier to read, we present Table 5, which displays MIC quantities supplied with the statistical significance quantitatives between compounds **9**–**11** and reference antibiotics **A1**–**A5**.

Analysis of the MIC and MBC values revealed the high bacteriostatic (Table 5) and bactericidal (Figure 9) activity of the compounds **9a**, **9b**, **9d**, **10a**, **10b**, **10d**, **11a**, **11b**, and **11d** solutions, which were superior to that of the standard antibiotics against the bacterial test systems used.

#### 2.2.6. Evaluation of Antioxidant Activity

The presence of a catechol moiety in the acylhydrazones’ structures allows for inference of their antioxidant properties. Therefore, we performed a coulometric evaluation of total antioxidant capacity using electrogenerated bromine as a reactive species. First, the stoichiometry of the reactions between acylhydrazones and the coulometric titrant was studied. The reactions proceeded quantitatively and quickly. The number of electrons participating in the reaction was calculated using Faraday’s formula (Table 6). The majority of acylhydrazones shows two-electron oxidation corresponding to the formation of *ortho*-quinone from the catechol moiety [70,71]. The presence of a pyridinium fragment in the structures of acylhydrazones **11a**–**d** leads to an increase in the number of electrons participating in the reaction with the electrogenerated bromine. In this case, three electrons are involved in the reaction, which is the result of bromine’s ability to participate in the electrophilic substitution in the aromatic ring [72].

Based on data obtained, the total antioxidant capacity of the studied acylhydrazones was evaluated (Table 6) and expressed as the amount of electricity used for titration and recalculated per mole of the target compound. Catechol and BHT were used as standards for the comparison of the data obtained.

All the acylhydrazones studied exhibited a statistically significant increase in total antioxidant capacity versus the reference compounds. These data agree well with the previously reported total antioxidant capacity for ammonium acylhydrazones containing a sterically hindered phenol fragment [16]. Other studies have also confirmed that phenolic *N*-acylhydrazones containing pyrogallol and/or catechol units [73], acylhydrazones derivatives bearing 5*H*-dibenzo[a,d][7]annulene and 4-hydroxy-3-methoxyphenyl moieties [74], and isatin acylhydrazones containing sterically hindered phenol moieties exhibit marked scavenging capacity toward the 2,2-diphenyl-1-picrylhydrazyl radical [75].

## 3. Materials and Methods

### 3.1. Chemistry

IR spectra were recorded using an IR Fourier spectrometer, namely, a Tensor 37 (Bruker Optik GmbH, Ettlingen, Germany), in the 400–3600 cm^−1^ range using KBr pellets. The infrared spectra of compounds **2** and **4** in the 4000–400 cm^−1^ range were recorded using an FSM 1201 Fourier-IR spectrometer in nujol. The ^1^H- and ^13^C-NMR spectra were recorded using a Bruker AVANCE 400 spectrometer (Bruker BioSpin, Rheinstetten, Germany) operating at 400 MHz (for ^1^H NMR) and 101 MHz (for ^13^C NMR), a Brucker spectrometer AVANCE*III*-500 (Bruker BioSpin, Rheinstetten, Germany) operating at 500 MHz (for ^1^H NMR) and 126 MHz (for ^13^C MMR), and a Bruker Avance 600 spectrometer (Bruker BioSpin, Rheinstetten, Germany) operating at 600 MHz (for ^1^H NMR) and 151 MHz (for ^13^C NMR). Chemical shifts were measured in δ (ppm) with reference to the solvent (δ = 2.50 ppm and 39.50 ppm for DMSO-*d*_6_, for ^1^H and ^13^C NMR, respectively. Elemental analysis was performed using a CHNS-O Elemental Analyser EuroEA3028-HT-OM (EuroVector S.p.A., Milan, Italy). The melting points were determined using a Stuart SMP10 apparatus (Stuart, Birmingham, UK).

Ammonium hydrazides **6**–**8** were prepared according to previously described methods [16,76,77].

**Synthesis of 2,3-dihydroxy-4,6-dialkylbenzaldehydes 2, 4 (general method).** 2,3-Dihydroxy-4,6-dialkylbenzaldehydes 2 and 4 were synthesized via the Duff reaction from corresponding 3,5-dialkylsubstituted catechols, similar to the one described earlier [78]. Thus, 0.02 mol of 3,5-dialkylsubstituted catechol and urotropine (2.8 g, 0.02 mol) were dissolved in acetic acid (50 mL) under Ar atmosphere and heated at 120 °C for 6 h. The reaction mixture was then cooled to 90 °C, and 50 mL of 40% aqueous H_2_SO_4_ solution was added. The precipitate was filtered and recrystallized from hexane or MeOH. The product appeared as yellow crystals.

**2,3-Dihydroxy-4,6-di-*tert*-pentylbenzaldehyde (2).** Yellow powder. Yield 48%, m.p. = 97–99 °C. IR spectrum, ν, cm^−1^: 1579 (C=C), 1621 (C=O), 3416 (OH). ^1^H NMR (400 MHz, CDCl_3_) δ 12.96 (s, 1H, OH), 10.69 (s, 1H, C(O)H), 6.75 (s, 1H, ArH), 6.01 (s, 1H, OH), 1.82 (q, *J* = 7.4 Hz, 4H, CH_2_), 1.88 (q, *J* = 7.4 Hz, 4H, CH_2_), *J* = 7.4 Hz), 1.37 (s, 6H, CH_3_), 1.44 (s, 6H, CH_3_), 0.67 (t, *J* = 7.4 Hz, 3H, CH_3_), 0.71 (t, *J* = 7.4 Hz, 3H, CH_3_). ^13^C NMR (101 MHz, CDCl_3_) δ 196.4 (CH), 151.7, 141.6, 141.2, 140.2, 118.0 (CH), 116.2, 39.5, 39.4, 38.2 (CH_2_), 32.9 (CH_2_), 31.3 (CH_3_), 27.0 (CH_3_), 9.38 (CH_3_), 9.35 (CH_3_). Found: C, 73.40; H, 9.45. Anal. calcd (%) for C_17_H_26_O_3_: C, 73.34; H, 9.41.

**2,3-Dihydroxy-4,6-bis(1-methylcyclohexyl)benzaldehyde (4).** Yellow powder. Yield 54%, m.p. = 90–92 °C. IR spectrum, ν, cm^−1^: 1569 (C=C), 1625 (C=O), 3435 (OH). ^1^H NMR (400 MHz, CDCl_3_) δ 12.97 (s, 1H, OH), 10.72 (s, 1H, C(O)H), 6.93 (s, 1H, ArH), 6.03 (s, 1H, OH), 2.26–2.16 (m, 2H, CH_2_), 2.00–1.90 (m, 2H, CH_2_), 1.84–1.74 (m, 2H, CH_2_), 1.72–1.50 (m, 10H, CH_2_), 1.48 (s, 3H, CH_3_), 1.47–1.35 (m, 4H, CH_2_), 1.33 (s, 3H, CH_3_). ^13^C NMR (101 MHz, CDCl_3_) δ 196.3 (CH), 152.2, 143.0, 141.6, 140.8, 116.9 (CH), 115.7, 41.2 (CH_2_), 39.4 (CH_2_), 38.9, 36.5, 29.1, 26.4 (CH_2_), 26.0 (CH_2_), 25.5, 22.9 (CH_3_), 22.7 (CH_3_). Found: C, 76.40; H, 9.20. Anal. calcd (%) for C_21_H_30_O_3_: C, 76.33; H, 9.15.

**Synthesis of ammonium acylhydrazones 9–11 (general method).** A corresponding amount of ammonium hydrazide (10 mmol) and one drop of trifluoroacetic acid were successively added to the solution of substituted catecholaldehyde (10 mmol) in 5 mL of absolute methanol. The reaction mixture was heated while stirring it at 60 °C for a corresponding amount of time (2–8.5 h). The progress of the reaction was monitored using TLC. The solvent was then rotary-evaporated. Diethyl ether was added to the residue and stirred for 15 min. The precipitate that formed was filtered off, washed with absolute ether, and dried in a vacuum.

The NMR spectra below show the characteristics of the major isomer.

**(2,3-Dihydroxy-4,6-diisopropylbenzilidenhydrazinocarbonylmethyl)-diethyl-methyl-ammonium bromide (9a)**. Pale yellow powder. Yield 94%, m.p. = 184–186 °C. IR spectrum, ν, cm^−1^: 1597 (C=C), 1716 (C=O), 2964 (CH), 3145 (NH), 3309 (OH). ^1^H NMR (400 MHz, DMSO-*d*_6_) δ 12.55 (br.s, 1H, OH), 12.05 (s, 1H, NH), 8.86 (s, 1H, =CH), 8.32 (s, 1H, OH), 6.64 (s, 1H, ArH), 4.21 (s, 2H, CH_2_C(O)), 3.61–3.55 (m, 4H, CH_2_N^+^), 3.27–3.23 (m, 1H, CH), 3.21–3.16 (m, 2H, CH), 3.19 (s, 3H, CH_3_N^+^), 1.31 (t, *J* = 7.2 Hz, 6H, CH_3_), 1.22 (d, *J* = 6.7 Hz, 6H, CH_3_), 1.18 (d, *J* = 6.9 Hz, 6H, CH_3_). ^13^C NMR (101 MHz, DMSO-*d*_6_) δ 159.6, 149.7 (CH), 147.2, 140.7, 138.8, 138.2, 113.2 (CH), 111.9, 58.5 (CH_2_C(O)), 58.1 (CH_2_N^+^), 48.4 (CH_3_N^+^), 28.3 (CH), 27.6 (CH), 24.5 (CH_3_), 22.6 (CH_3_), 8.3 (CH_3_). Found: C, 54.00; H, 7.67; Br, 17.90; N, 9.40. Anal. calcd (%) for C_20_H_34_BrN_3_O_3_: C, 54.05; H, 7.71; Br, 17.98; N, 9.46.

**1-[4,6-Bis-(1,1-dimethylpropyl)-2,3-dihydroxybenzylidenehydrazinocarbonylmethyl]-diethyl-methyl-ammonium bromide (9b)**. Pale yellow powder. Yield 92%, m.p. = 202–204 °C. IR spectrum, ν, cm^−1^: 1571 (C=C), 1700 (C=O), 2962 (CH), 3153 (NH), 3403 (OH). ^1^H NMR (600 MHz, DMSO-*d*_6_) δ 12.59 (br.s, 1H, OH), 12.54 (s, 1H, NH), 9.16 (s, 1H, =CH), 8.23 (s, 1H, OH), 6.65 (s, 1H, ArH), 4.21 (s, 2H, CH_2_C(O)), 3.62–3.55 (m, 4H, CH_2_N^+^), 3.19 (s, 3H, CH_3_N^+^), 1.83 (q, *J* = 7.4 Hz, 2H, CH_2_), 1.74 (q, *J* = 7.5 Hz, 2H, CH_2_), 1.35 (s, 6H, CH_3_), 1.31–1.28 (m, 12H, CH_3_), 0.62–0.58 (m, 6H, CH_3_). ^13^C NMR (151 MHz, DMSO-*d*_6_) δ 159.7, 151.2 (CH), 148.4, 142.6, 137.3, 135.3, 117.7 (CH), 112.6, 58.5 (CH_2_C(O)), 58.1 (CH_2_N^+^), 48.4 (CH_3_N^+^), 39.3, 38.9, 36.4 (CH_2_), 32.9 (CH_2_), 31.0 (CH_3_), 27.6 (CH_3_), 9.84 (CH_3_), 9.77 (CH_3_), 8.3 (CH_3_). Found: C, 57.50; H, 8.35; Br, 15.90; N, 8.35. Anal. calcd (%) for C_24_H_42_BrN_3_O_3_: C, 57.59; H, 8.46; Br, 15.96; N, 8.40.

**1-(4,6-Dicyclohexyl-2,3-dihydroxybenzylidenehydrazinocabonylmethyl)-diethyl-methyl-ammonium bromide (9c)**. Pale yellow powder. Yield 88%, m.p. = 211–213 °C. IR spectrum, ν, cm^−1^: 1571 (C=C), 1686 (C=O), 2926 (CH), 3151 (NH), 3392 (OH). ^1^H NMR (400 MHz, DMSO-*d*_6_) δ 12.62 (br.s, 1H, OH), 12.04 (s, 1H, NH), 8.89 (s, 1H, =CH), 8.31 (s, 1H, OH), 6.60 (s, 1H, ArH), 4.24 (s, 2H, CH_2_C(O)), 3.61–3.56 (m, 4H, CH_2_N^+^), 3.20 (s, 3H, CH_3_N^+^), 2.92–2.87 (m, 1H, CH), 2.75–2.71 (m, 1H, CH), 1.80–1.71 (m, 10H, CH_2_), 1.31 (t, *J* = 7.3 Hz, 6H, CH_3_), 1.42–1.25 (m, 10H, CH_2_). ^13^C NMR (101 MHz, DMSO-*d*_6_) δ 159.5, 149.6 (CH), 147.1, 140.7, 138.2, 137.3, 114.3 (CH), 111.9, 58.5 (CH_2_C(O)), 58.1 (CH_2_N^+^), 48.4 (CH_3_N^+^), 38.8 (CH), 37.7 (CH), 34.8 (CH_2_), 32.7 (CH_2_), 27.1 (CH_2_), 27.0 (CH_2_), 26.3 (CH_2_), 26.1 (CH_2_), 8.3 (CH_3_). Found: C, 59.50; H, 8.01; Br, 15.17; N, 7.99. Anal. calcd (%) for C_26_H_42_BrN_3_O_3_: C, 59.53; H, 8.07; Br, 15.23; N, 8.01.

**[2,3-Dihydroxy-4,6-bis-(1-methylcyclohexyl)-benzylidenehydrazinocarbonylmethyl]-diethyl-methyl-ammonium bromide (9d)**. Pale yellow powder. Yield 83%, m.p. = 210–211 °C. IR spectrum, ν, cm^−1^: 1570 (C=C), 1692 (C=O), 2929 (CH), 3148 (NH), 3418 (OH). ^1^H NMR (400 MHz, DMSO-*d*_6_) δ 12.61 (s, 1H, NH), 12.53 (br.s, 1H, OH), 9.17 (s, 1H, =CH), 8.25 (s, 1H, OH), 6.82 (s, 1H, ArH), 4.21 (s, 2H, CH_2_C(O)), 3.62- 3.55 (m, 4H, CH_2_N^+^), 3.20 (s, 3H, CH_3_N^+^), 2.20–2.15 (m, 2H, CH_2_), 1.88–1.82 (m, 2H, CH_2_), 1.76–1.71 (m, 2H, CH_2_), 1.66–1.40 (m, 14H, CH_2_), 1.37 (s, 3H, CH_3_), 1.31 (t, *J* = 7.3 Hz, 6H, CH_3_), 1.28 (s, 3H, CH_3_). ^13^C NMR (101 MHz, DMSO-*d*_6_) δ 159.7, 151.2 (CH), 149.0, 142.5, 139.1, 136.1, 116.4 (CH), 112.0, 58.5 (CH_2_C(O)), 58.1 (CH_2_N^+^), 48.4 (CH_3_N^+^), 38.8, 38.6, 36.6 (CH_2_), 28.1 (CH_3_), 26.5 (CH_2_), 26.1 (CH_2_), 25.9 (CH_3_), 22.9 (CH_2_), 22.8 (CH_2_), 8.3 (CH_3_). Found: C, 60.80; H, 8.31; Br, 14.39; N, 7.50. Anal. calcd (%) for C_28_H_46_BrN_3_O_3_: C, 60.86; H, 8.39; Br, 14.46; N, 7.60.

**(4,6-Diadamantan-1-yl-2,3-dihydroxybenzylidenehydrazinocarbonylmethyl)-diethyl-methyl-ammonium bromide (9e)**. Pale yellow powder. Yield 84%, m.p. = 290 °C (decomp.). IR spectrum, ν, cm^−1^: 1572 (C=C), 1696 (C=O), 2904 (CH), 3150 (NH), 3379 (OH), 3509 (OH). ^1^H NMR (600 MHz, DMSO-*d*_6_) δ 12.76 (br.s, 1H, OH), 12.62 (s, 1H, NH), 9.33 (s, 1H, =CH), 8.23 (s, 1H, OH), 6.69 (s, 1H, ArH), 4.24 (s, 2H, CH_2_C(O)), 3.62–3.58 (m, 4H, CH_2_N^+^), 3.20 (s, 3H, CH_3_N^+^), 2.14–1.70 (m, 30H, Ad), 1.31 (t, *J* = 7.3 Hz, 6H, CH_3_). ^13^C NMR (151 MHz, DMSO-*d*_6_) δ 160.0, 150.7 (CH), 148.7, 142.8, 139.6, 137.2, 114.5 (CH), 112.2, 58.4 (CH_2_C(O)), 58.1 (CH_2_N^+^), 48.4 (CH_3_N^+^), 43.7 (CH_2_), 40.0 (CH_2_), 37.9, 37.5, 37.1 (CH_2_), 36.5 (CH_2_), 29.2 (CH), 28.9 (CH), 8.3 (CH_3_). Found: C, 64.96; H, 8.02; Br, 12.71; N, 6.68. Anal. calcd (%) for C_34_H_50_BrN_3_O_3_: C, 64.90; H, 7.95; Br, 12.68; N, 6.60.

**1-(2,3-Dihydroxy-4,6-diisopropylbenzilidenehydrazinocarbonylmethyl)-4-aza-1-azonia-bicyclo[2.2.2]octane bromide (10a)**. Pale yellow powder. Yield 67%, m.p. = 152–153 °C. IR spectrum, ν, cm^−1^: 1601 (C=C), 1691 (C=O), 2962 (CH), 3145 (NH), 3412 (OH). ^1^H NMR (500 MHz, DMSO-*d*_6_) δ 12.65 (br.s, 1H, OH), 12.05 (s, 1H, NH), 8.88 (s, 1H, =CH), 8.35 (s, 1H, OH), 6.65 (s, 1H, ArH), 4.22 (s, 2H, CH_2_C(O)), 3.59 (t, *J* = 7.2 Hz, 6H, CH_2_N^+^), 3.27–3.21 (m, 1H, CH), 3.20–3.16 (m, 1H, CH), 3.11 (t, *J* = 7.2 Hz, 6H, CH_2_), 1.22 (d, *J* = 6.5 Hz, 6H, CH_3_), 1.18 (d, *J* = 6.9 Hz, 6H, CH_3_). ^13^C NMR (126 MHz, DMSO-*d*_6_) δ 159.1, 149.8 (CH), 147.1, 140.7, 138.9, 138.2, 113.2 (CH), 112.0, 62.0 (CH_2_C(O)), 53.3 (CH_2_N^+^), 45.1 (CH_2_), 28.3 (CH), 27.6 (CH), 24.5 (CH_3_), 22.6 (CH_3_). Found: C, 53.65; H, 7.02; Br, 16.93; N, 11.00. Anal. calcd (%) for C_21_H_33_BrN_4_O_3_: C, 53.73; H, 7.09; Br, 17.02; N, 11.04.

**1-[4,6-Bis-(1,1-dimethylpropyl)-2,3-dihydroxybenzylidenehydrazinocarbonylmethyl]-4-aza-1-azonia-bicyclo[2.2.2]octane bromide (10b).** Pale yellow powder. Yield 76%, m.p. = 250–252 °C. IR spectrum, ν, cm^−1^: 1561 (C=C), 1686 (C=O), 2966 (CH), 3152 (NH), 3410 (OH). ^1^H NMR (400 MHz, DMSO-*d*_6_) δ 12.68 (br.s, 1H, OH), 12.55 (s, 1H, NH), 9.19 (s, 1H, =CH), 8.25 (s, 1H, OH), 6.66 (s, 1H, ArH), 4.24 (s, 2H, CH_2_C(O)), 3.61 (t, *J* = 7.6 Hz, 6H, CH_2_N^+^), 3.12 (t, *J* = 7.6 Hz, 6H, CH_2_), 1.83 (q, *J* = 7.4 Hz, 2H, CH_2_), 1.76 (q, *J* = 7.6 Hz, 2H, CH_2_), 1.36 (s, 6H, CH_3_), 1.31 (s, 6H, CH_3_), 0.63–0.58 (m, 6H, CH_3_). ^13^C NMR (101 MHz, DMSO-*d*_6_) δ 159.1, 151.2 (CH), 148.4, 142.5, 137.3, 135.3, 117.7 (CH), 112.6, 61.9 (CH_2_C(O)), 53.3 (CH_2_N^+^), 45.0 (CH_2_), 39.2, 38.9, 36.3 (CH_2_), 32.8 (CH_2_), 30.9 (CH_2_), 27.6 (CH_2_), 9.8 (CH_3_). Found: C, 57.07; H, 7.80; Br, 15.11; N, 10.60. Anal. calcd (%) for C_25_H_41_BrN_4_O_3_: C, 57.14; H, 7.86; Br, 15.20; N, 10.66.

**1-(4,6-Dicyclohexyl-2,3-dihydroxybenzylidenehydrazinocabonylmethyl)-4-aza-1-azonia-bicyclo[2.2.2]octane bromide (10c)**. Pale yellow powder. Yield 82%, m.p. = 253–254 °C. IR spectrum, ν, cm^−1^: 1597 (C=C), 1676 (C=O), 2924 (CH), 3052 (ArH), 3145 (NH), 3408 (OH), 3500 (OH). ^1^H NMR (500 MHz, DMSO-*d*_6_) δ 12.51 (br.s, 1H, OH), 12.02 (s, 1H, NH), 8.83 (s, 1H, =CH), 8.31 (s, 1H, OH), 6.60 (s, 1H, ArH), 4.20 (s, 2H, CH_2_C(O)), 3.58 (t, *J* = 7.3 Hz, 6H, CH_2_N^+^), 3.10 (t, *J* = 7.3 Hz, 6H, CH_2_), 2.91–2.87 (m, 1H, CH), 2.74–2.71 (m, 1H, CH), 1.81–1.71 (m, 10H, CH_2_), 1.47–1.24 (m, 10H, CH_2_). ^13^C NMR (126 MHz, DMSO-*d*_6_) δ 159.0, 149.5 (CH), 147.0, 140.6, 138.1, 137.3, 114.3 (CH), 111.9, 62.0 (CH_2_C(O)), 53.4 (CH_2_N^+^), 45.0 (CH_2_), 38.9 (CH), 37.1 (CH), 34.8 (CH_2_), 32.6 (CH_2_), 27.1 (CH_2_), 27.0 (CH_2_), 26.2 (CH_2_), 26.1 (CH_2_). Found: C, 58.95; H, 7.47; Br, 14.43; N, 10.11. Anal. calcd (%) for C_27_H_41_BrN_4_O_3_: C, 59.01; H, 7.52; Br, 14.54; N, 10.20.

**[2,3-Dihydroxy-4,6-bis-(1-methylcyclohexyl)-benzylidenehydrazinocarbonylmethyl]-1,4-diazabicyclo[2.2.2]octan-1-ium bromide (10d).** Pale yellow powder. Yield 89%, m.p. = 243–245 °C. IR spectrum, ν, cm^−1^: 1584 (C=C), 1677 (C=O), 2927 (CH), 3145 (NH), 3432 (OH). ^1^H NMR (400 MHz, DMSO-*d*_6_) δ 12.58 (br.s, 1H, OH), 12.56 (s, 1H, NH), 9.19 (s, 1H, =CH), 8.26 (s, 1H, OH), 6.83 (s, 1H, ArH), 4.21 (s, 2H, CH_2_C(O)), 3.60 (t, *J* = 7.6 Hz, 6H, CH_2_N^+^), 3.11 (t, *J* = 7.6 Hz, 6H, CH_2_), 2.19–2.14 (m, 20H, CH_2_), 1.88–1.82 (m, 20H, CH_2_), 1.76–1.71 (m, 20H, CH_2_), 1.66–1.42 (m, 20H, CH_2_), 1.37 (s, 3H, CH_3_), 1.28 (s, 3H, CH_3_). ^13^C NMR (101 MHz, DMSO-*d*_6_) δ 159.2, 151.3 (CH), 148.9, 142.5, 139.2, 136.2, 116.5 (CH), 112.1, 61.9 (CH_2_C(O)), 53.3 (CH_2_N^+^), 45.0 (CH_2_), 40.2 (CH_2_), 38.8, 38.6, 36.6 (CH_2_), 28.1 (CH_3_), 26.5 (CH_3_), 26.1 (CH_2_), 25.9 (CH_2_), 22.9 (CH_2_), 22.8 (CH_2_). Found: C, 60.21; H, 7.80; Br, 13.72; N, 9.61. Anal. calcd (%) for C_29_H_45_BrN_4_O_3_: C, 60.30; H, 7.85; Br, 13.83; N, 9.70.

**1-(4,6-Diadamantan-1-yl-2,3-dihydroxybenzylidenehydrazinocarbonylmethyl)-4-aza-1-azonia-bicyclo[2.2.2]octane bromide (10e)**. Pale yellow powder. Yield 84%, m.p. = 298 °C (decomp.). IR spectrum, ν, cm^−1^: 1547 (C=C), 1699 (C=O), 2904 (CH), 3106 (NH), 3485 (OH). ^1^H NMR (400 MHz, DMSO-*d*_6_) δ 12.72–12.70 (m, 2H, OH, NH), 9.37 (s, 1H, =CH), 8.25 (s, 1H, OH), 6.70 (s, 1H, ArH), 4.28 (s, 2H, CH_2_C(O)), 3.61 (t, *J* = 7.3 Hz, 6H, CH_2_N^+^), 3.11 (t, *J* = 7.3 Hz, 6H, CH_2_), 2.09–1.73 (m, 30H, Ad). ^13^C NMR (101 MHz, DMSO-*d*_6_) δ 159.4, 150.8 (CH), 148.7, 142.8, 139.7, 137.2, 114.5 (CH), 112.1, 61.7 (CH_2_C(O)), 53.3 (CH_2_N^+^), 45.0 (CH_2_), 43.7 (CH_2_), 40.1 (CH_2_), 37.9, 37.5, 37.1 (CH_2_), 36.5 (CH_2_), 29.2 (CH), 28.8 (CH). Found: C, 64.25; H, 7.50; Br, 12.13; N, 8.50. Anal. calcd (%) for C_35_H_49_BrN_4_O_3_: C, 64.31; H, 7.56; Br, 12.22; N, 8.57.

**1-(2,3-Dihydroxy-4,6-diisopropylbenzilidenehydrazinocarbonylmethyl)-2,4-dimethylpyridinium bromide (11a)**. Pale yellow powder. Yield 94%, m.p. = 188–189 °C. IR spectrum, ν, cm^−1^: 1561 (C=C), 1701 (C=O), 2961 (CH), 3245 (NH), 3421 (OH). ^1^H NMR (400 MHz, DMSO-*d*_6_) δ 12.54 (br.s, 1H, OH), 11.98 (s, 1H, NH), 8.89 (br.s, 2H, ArH, =CH), 8.48 (d, *J* = 7.9 Hz, 1H, ArH), 8.33 (br.s, 1H, OH), 8.00–7.91 (m, 1H, ArH), 6.65 (s, 1H, ArH), 5.67 (s, 2H, CH_2_C(O)), 3.31–3.14 (m, 2H, CH), 2.68 (s, 3H, CH_3_), 2.54 (s, 3H, CH_3_), 1.23 (d, *J* = 6.6 Hz, 6H, CH_3_), 1.17 (d, *J* = 6.7 Hz, 6H, CH_3_). ^13^C NMR (101 MHz, DMSO-*d*_6_) δ 160.9, 155.9, 149.4 (CH), 147.0 (CH), 145.3 (CH), 140.7, 139.2, 138.8, 138.1, 124.9 (CH), 113.2 (CH), 112.0, 59.4 (CH_2_N^+^), 28.4 (CH), 27.6 (CH), 24.5 (CH_3_), 22.6 (CH_3_), 19.7 (CH_3_), 17.5 (CH_3_). Found: C, 56.83; H, 6.43; Br, 17.15; N, 9.00. Anal. calcd (%) for C_22_H_30_BrN_3_O_3_: C, 56.90; H, 6.51; Br, 17.21; N, 9.05.

**1-[4,6-Bis-(1,1-dimethylpropyl)-2,3-dihydroxybenzylidenehydrazinocarbonylmethyl]-2,3-dimethylpyridinium bromide (11b)**. Pale yellow powder. Yield 93%, m.p. = 207–208 °C. IR spectrum, ν, cm^−1^: 1570 (C=C), 1707 (C=O), 2963 (CH), 3150 (NH), 3401 (OH). ^1^H NMR (500 MHz, DMSO-*d*_6_) δ 12.66 (br.s, 1H, OH), 12.45 (s, 1H, NH), 9.19 (s, 1H, =CH), 8.89 (d, *J* = 6.2 Hz, 1H, ArH), 8.48 (d, *J* = 7.6 Hz, 1H, ArH), 8.21 (s, 1H, OH), 7.94 (t, *J* = 7.3 Hz, 1H, ArH), 6.66 (s, 1H, ArH), 5.68 (s, 2H, CH_2_C(O)), 2.68 (s, 3H, CH_3_), 2.53 (s, 3H, CH_3_), 1.86–1.73 (m, 4H, CH_2_), 1.37 (s, 6H, CH_3_), 1.30 (s, 6H, CH_3_), 0.65–0.57 (m, 6H, CH_3_). ^13^C NMR (126 MHz, DMSO-*d*_6_) δ 160.9, 156.1, 150.9 (CH), 148.3, 147.0 (CH), 145.3 (CH), 142.5, 138.8, 137.2, 135.2, 124.9 (CH), 117.7 (CH), 112.7, 59.3 (CH_2_N^+^), 39.3, 38.9, 36.4 (CH_2_), 32.8 (CH_2_), 30.9 (CH_3_), 27.6 (CH_3_), 19.8 (CH_3_), 17.5 (CH_3_), 9.83 (CH_3_), 9.79 (CH_3_). Found: C, 59.91; H, 7.30; Br, 15.27; N, 8.01. Anal. calcd (%) for C_26_H_38_BrN_3_O_3_: C, 60.00; H, 7.36; Br, 15.35; N, 8.07.

**1-(4,6-Dicyclohexyl-2,3-dihydroxybenzylidenehydrazinocabonylmethyl)-2,3-dimethylpyridinium bromide (11c)**. Pale yellow powder. Yield 98%, m.p. = 153–155 °C. IR spectrum, ν, cm^−1^: 1598 (C=C), 1696 (C=O), 2925 (CH), 3120 (NH), 3410 (OH). ^1^H NMR (400 MHz, DMSO-*d*_6_) δ 12.70 (s, 1H, OH), 11.97 (s, 1H, NH), 8.94–8.89 (m, 1H, ArH), 8.92 (s, 1H, =CH), 8.48 (d, *J* = 8.0 Hz, 1H, ArH), 8.30 (s, 1H, OH), 8.00–7.91 (m, 1H, ArH), 6.60 (s, 1H, ArH), 5.71 (s, 2H, CH_2_C(O)), 2.95–2.85 (m, 1H, CH), 2.84–2.71 (m, 1H, CH), 2.68 (s, 3H, CH_3_), 2.53 (s, 3H, CH_3_), 1.86–1.64 (m, 10H, CH_2_), 1.49–1.18 (m, 10H, CH_2_). ^13^C NMR (101 MHz, DMSO-*d*_6_) δ 165.0, 160.9, 155.9, 149.3 (CH), 146.8 (CH), 145.4 (CH), 140.7, 138.7, 138.1, 137.3, 124.9 (CH), 114.3 (CH), 112.0, 59.3 (CH_2_N^+^), 38.9 (CH), 37.7 (CH), 34.8 (CH_2_), 32.7 (CH_2_), 27.1 (CH_2_), 27.0 (CH_2_), 26.3 (CH_2_), 26.2 (CH_2_), 19.7 (CH_3_), 17.5 (CH_3_). Found: C, 61.69; H, 6.95; Br, 14.60; N, 7.63. Anal. calcd (%) for C_28_H_38_BrN_3_O_3_: C, 61.76; H, 7.03; Br, 14.67; N, 7.72.

**1-[2,3-Dihydroxy-4,6-bis-(1-methylcyclohexyl)-benzylidenehydrazinocarbonylmethyl]-2,3-dimethylpyridinium bromide (11d)**. Pale yellow powder. Yield 98%, m.p. = 158–160 °C. IR spectrum, ν, cm^−1^: 1570 (C=C), 1700 (C=O), 2926 (CH), 3145 (NH), 3412 (OH). ^1^H NMR (600 MHz, DMSO-*d*_6_) δ 12.65 (s, 1H, OH), 12.48 (s, 1H, NH), 9.21 (s, 1H, =CH), 8.89 (d, *J* = 6.3 Hz, 1H, ArH), 8.48 (d, *J* = 7.7 Hz, 1H, ArH), 8.22 (s, 1H, OH), 7.96–7.93 (m, 1H, ArH), 6.82 (s, 1H, ArH), 5.68 (s, 2H, CH_2_C(O)), 2.68 (s, 3H, CH_3_), 2.53 (s, 3H, CH_3_), 2.16–2.14 (m, 2H, CH_2_), 1.87–1.84 (m, 2H, CH_2_), 1.75–1.74 (m, 2H, CH_2_), 1.62–1.48 (m, 14H, CH_2_), 1.38 (s, 3H, CH_3_), 1.26 (s, 3H, CH_3_). ^13^C NMR (151 MHz, DMSO-*d*_6_) δ 160.9, 156.1, 151.0 (CH), 148.8, 147.0 (CH), 145.3 (CH), 142.5, 139.1, 138.8, 136.1, 124.9 (CH), 116.5 (CH), 112.2, 59.3 (CH_2_N^+^), 38.8, 38.7, 36.6 (CH_2_), 28.1 (CH_3_), 26.5 (CH_2_), 26.1 (CH_2_), 25.9 (CH_3_), 23.0 (CH_2_), 22.9 (CH_2_), 19.8 (CH_3_), 17.5 (CH_3_). Found: C, 62.87; H, 7.30; Br, 13.87; N, 7.29. Anal. calcd (%) for C_30_H_42_BrN_3_O_3_: C, 62.93; H, 7.39; Br, 13.95; N, 7.34.

**1-(4,6-Diadamantan-1-yl-2,3-dihydroxybenzylidenehydrazinocarbonylmethyl)-2,3-dimethylpyridinium bromide (11e)**. Pale yellow powder. Yield 72%, m.p. = 286 °C (decomp.). IR spectrum, ν, cm^−1^: 1561 (C=C), 1694 (C=O), 2904 (CH), 3130 (NH), 3404 (OH). ^1^H NMR (400 MHz, DMSO-*d*_6_) δ 12.80 (br.s, 1H, OH), 12.66 (s, 1H, NH), 9.39 (s, 1H, =CH), 8.92 (d, *J* = 6.2 Hz, 1H, ArH), 8.49 (d, *J* = 7.7 Hz, 1H, ArH), 8.23 (s, 1H, OH), 7.98–7.95 (m, 1H, ArH), 6.70 (s, 1H, ArH), 5.73 (s, 2H, CH_2_C(O)), 2.69 (s, 3H, CH_3_), 2.54 (s, 3H, CH_3_), 2.13–1.70 (m, 30H, Ad). ^13^C NMR (101 MHz, DMSO-*d*_6_) δ 161.6, 156.6, 150.9 (CH), 149.2, 147.5 (CH), 145.9 (CH), 143.3, 140.1, 139.3, 137.6, 125.4 (CH), 115.0 (CH), 112.8, 60.0 (CH_2_N^+^), 44.2 (CH_2_), 41.0 (CH_2_), 38.4, 38.0, 37.6 (CH_2_), 37.03 (CH_2_), 36.95 (CH_2_), 29.7 (CH), 29.4 (CH), 20.3 (CH_3_), 18.1 (CH_3_). Found: C, 66.59; H, 7.07; Br, 12.25; N, 6.40. Anal. calcd (%) for C_36_H_46_BrN_3_O_3_: C, 66.66; H, 7.15; Br, 12.32; N, 6.48.

### 3.2. Biological Studies

#### 3.2.1. Antimicrobial Activity

Gram-positive bacteria (*Staphylococcus aureus* ATCC 6538P FDA 209P, *Bacillus cereus* ATCC 10702 NCTC 8035, and the methicillin-resistant *Staphylococcus aureus* strains MRSA-1 and MRSA-2) were used as test objects. We used norfloxacin as a reference drug for studying antibacterial activity. Bacteriostatic and fungistatic properties were studied via serial dilutions in liquid nutrient media using the procedures described in [79], determining the MIC, the concentration at which the growth and production of the test microorganism are inhibited. The bactericidal (MBC) activity, causing complete death of the pathogen, was determined according to the previously described procedure [80].

#### 3.2.2. Crystal Violet (CV) Assay

The change in the absorption of the CV dye by *Staphylococcus aureus* cells grown on Muller–Hinton agar was determined as described in [81]. The cultured and washed cells were resuspended in phosphate buffer solution to obtain 2 × 10^8^ cfu/mL. The resulting inoculum was added to the test compounds in a ratio of 1:1 and incubated for 30 min at 37 °C. CV dye was then added to the solution, the concentration of which was 0.001% in the run. The optical density (OD) of the supernatant was determined at a wavelength of 540 nm using an Invitrologic microplate reader (Novosibirsk, Russia). The percentage of CV uptake was calculated using the following formula:$$CV=\frac{OD_{sample}-OD_{control}}{OD_{CV\;solution}}\times 100,$$
where *OD*_sample_, *OD*_control_, and *OD*_solution_ denote the values of optical density of the sample, the control (without the test compound), and the CV solutions (without cells), respectively.

#### 3.2.3. Cytotoxic Activity

##### Cell Lines and Their Cultivation

In the experiments, we used the cell culture WI-38 VA-13 subline 2RA (human embryonic lung) from the collection of type cultures of the Institute of Cytology of the Russian Academy of Sciences. The cells were grown in Eagle’s nutrient medium (PanEco, Moscow, Russia), containing fetal bovine serum (10% by volume) (BIOSERA, Cholet, France) and 1% non-essential amino acids (1% by volume) (PanEco, Moscow, Russia). Cultivation was carried out at 37 °C in a humidified CO_2_ atmosphere (5%).

##### Determination of Cell Viability

Cell viability was determined using the MTT test. Cells were seeded in a 96-well plate at 1 × 10^3^ cells/100 µL of the complete nutrient medium and cultured at 37 °C in CO_2_ (5%). After 24 h of incubation, various concentrations of test compounds in the range of 0.1 to 100 μM were added to the cell cultures, and then the cells were cultured under the same conditions for 24 h. For each concentration, the experiment was carried out in triplicate.

All compounds were dissolved in DMSO and then diluted with the medium to the required concentration. The final content of DMSO in the well did not exceed 1%, nor did it have a toxic effect on the cells. DMSO was also added to the control wells in a volume of 1%. After the incubation period, MTT (3-(4,5-dimethylthiazol-2-yl)-2,5-diphenyltetrazolium bromide (final concentration was 0.5 mg/mL) was added to each well, and the plates used were additionally incubated for 4 h until the characteristic color appeared.

Optical density was recorded at 540 nm using an Invitrologic plate reader (Novosibirsk, Russia). IC50 (half maximum inhibitory concentration) was calculated using an online tool: MLA—“Quest Graph™ IC50 Calculator”. AAT Bioquest, Inc., Pleasanton, CA, USA, https://www.aatbio.com/tools/ic50-calculator (accessed on 11 March 2024). The experiments were repeated three times.

#### 3.2.4. Hemolytic Activity

The hemolytic activity of the compounds analyzed was evaluated according to a previously reported method [82] by comparing the optical density of the solution containing the test compound with the optical density of a human blood solution at 100% hemolysis measured using a microplate reader Invitrologic plate reader (Novosibirsk, Russia) at λ = 540 nm. A control sample corresponding to zero hemolysis (negative control) was prepared by adding 10% erythrocyte suspension to saline (0.9% NaCl). A control sample corresponding to 100% hemolysis was prepared by adding 10% erythrocyte suspension to distilled water (positive control).

#### 3.2.5. Anticoagulant and Antiplatelet Activity Study

The in vitro experiments were performed using the blood of healthy male donors aged 18–24 years (amounting to a total of 56 donors). This study was approved by the Ethics Committee of the Federal State Budgetary Educational Institution of Higher Education at Bashkir State Medical University of the Ministry of Health of the Russian Federation (No. 2, dated 12 November 2020). We obtained informed consent from all the participants before sampling their blood. Blood was collected from the cubital vein using the BD Vacutainer^®^ vacuum blood collection system (Becton, Dickinson, and Company, Franklin Lakes, NJ, USA). A 3.8% sodium citrate solution in a 9:1 ratio was used as a venous blood stabilizer. The study of the effect on platelet aggregation was performed via the Born method [83] using an aggregometer AT-02 (SPC Medtech, Moscow, Russia). The assessment of the antiplatelet activity of the studied compounds and reference preparations started with a final concentration of 2 × 10^−3^ mol/L (corresponding to the therapeutic concentration and half the toxic concentration when administering acetylsalicylic acid orally) [84]. Adenosine diphosphate (ADP; 20 μg/mL) and collagen (5 mg/mL)—manufactured by Tehnologia-Standart Company, Barnaul, Russia—were used as inducers of aggregation. The study on anticoagulant activity was performed by conducting standard recognized clotting tests using an optical two-channel automatic analyzer of blood coagulation: Solar CGL 2110 (CJSC SOLAR, Minsk, Belarus). The following parameters were studied: activated partial thromboplastin time (APTT), prothrombin time (PT), and fibrinogen concentrations measured according to the Clauss method. The anticoagulant activity of the studied compounds and reference preparation was determined at a concentration of 0.5 g/L using the reagents manufactured by Tehnologia-Standart Company (Barnaul, Russia). The results of this study were processed using the statistical package Statistica 10.0 (StatSoft Inc., Tulsa, OK, USA). The Shapiro–Wilk’s test was used to check the normality of the actual dataset distribution. The form of the distribution of the data obtained differed from the normal one; therefore, non-parametric methods were used for further analysis. The data were presented as medians and 25 and 75 percentiles. An analysis of variance was conducted using the Kruskal–Wallis test. A *p*-value of 0.05 was considered statistically significant.

#### 3.2.6. Antiphytopathogenic Activity

Strains of the bacterial phytopathogens *Micrococcus luteus* B-109, *Pectobacterium atrosepticum* 1043, *Pectobacterium carotovorum* subsp. *carotovorum* MI, *Pseudomonas fluorescens* EL-2.1, and *Xanthomonas campestris* B-610 were obtained from the Collection of Rhizosphere Microorganisms, IBPPM RAS (WFCC no. 975, WDCM no. 1021) (CM IBPPM, http://collection.ibppm.ru (accessed on 13 May 2025)). The bacteria *M. luteus*, *P. carotovorum* subsp. *carotovorum*, *P. atrosepticum*, and *Ps. fluorescens* were grown in meat peptone medium (BP), and *X. campestris* was grown in a medium containing glucose, yeast extract, and calcium carbonate (GYCa). The solid media contained Bacto agar (18 g/L); the pH was adjusted to 7.2–7.4. All bacterial cultures were grown at 28 °C.

The antibacterial activity of the test compounds and **9a**–**e**, **10a**–**e**, and **11a**–**e** was explored using the agar well diffusion method by measuring the diameter of the growth inhibition zones to determine the compounds’ bactericidal activity. In this stage, 6 mm wells were made in agar medium (GYCa for *Xanthomonas campestris* or BP for other bacteria). Bacterial suspensions were distributed over the agar surface, and the solution containing the tested compound (150 μL) was added to each well. The width of the growth inhibition zones around the wells was determined after incubation for 36–40 h. The bacteriostatic and bactericidal activities of the compounds against phytopathogenic bacteria were examined, also referring to the typical method used to determine the minimal inhibitory concentration (MIC) and minimal bactericidal concentration (MBC). The MIC was defined as the lowest concentration of chemicals that inhibited bacterial growth in nutrient liquid. The minimum bactericidal or lethal concentration (MBC) was defined, based on the viability of the strain on a solid medium, as the lowest concentration of the tested compound that yielded no colony growth [85].

The MICs of the tested samples were determined using a broth dilution method, with visible bacterial growth observed in the test-tubes [86]. The stock solutions of the compounds analyzed or standard antibiotics (500 µg/mL) were added to 2 mL of nutrient liquid medium, and serial dilution was performed to produce chemical-containing media with a final compound concentration within the range of 0.16 to 50 µg/mL. Then, a bacterial suspension (30 microl, OD_600_ = 0.5) was added to each test-tube and incubated for 24 h at 28 °C. The dilution tubes were then tested for the absence or presence of visible growth in comparison with that in the negative controls. The endpoint for MIC was the lowest concentration of the compound at which there was no visible growth.

To determine the MBC values, we carried out serial 2-fold dilutions of the test substances in a nutrient medium, incubated with the bacteria for 24 h, and used as the inoculum suspensions to be spotted on agar-filled Petri dishes. The inoculated plates were incubated at 28 °C for 24 h. The subculturing of broth from tubes containing the MIC concentration and above (from each clear tube) onto agar plates allowed us to define the minimum microbicidal concentrations [87]. MBC values were recorded by reading the lowest compound concentration that inhibited visible solid-phase bacterial growth. Growth (negative) controls were analyzed on both liquid and agar media of the same composition that did not contain the chemicals analyzed, and the uninoculated broth was maintained as a sterility control. The assays were carried out in triplicate.

#### 3.2.7. Coulometric Evaluation of Total Antioxidant Capacity

Catechol and BHT of 99% purity were purchased from Aldrich (Steinheim, Germany). We prepared 1.0–10 mM stock solutions of synthesized acylhydrazones, catechol, and butylated hydroxytoluene by dissolving an exact amount of the substance in ethanol (rectificate).

The total antioxidant capacity of the compounds was studied using coulometric titration with electrogenerated bromine [72]. In brief, bromine was electrochemically generated under constant-current coulometry conditions from 0.2 mol L^−1^ KBr in 0.1 M sulfuric acid (precursor of bromine) using a “Exper-006” coulometric analyzer (Econix-Expert, Moscow, Russia). The four-electrode system employed consisted of a bare platinum wire with a surface area of 0.5 cm^2^ as a working electrode, platinum wire separated from the anodic compartment of the cell with the semipermeable membrane as an auxiliary electrode, and two polarized platinum needles as indicator electrodes (Δ*E* = 200 mV) for the detection of the titration end point.

Electrolysis was performed in a 50 mL glass cell containing 20 mL of bromine precursor at a current density of 5 mA cm^−2^. After switching on the generating circuit and achieving a 40 µA indicator current, we added an aliquot (50–150 μL) of the compound standard solution in ethanol (rectificate) to the cell and simultaneously started the timer. The titration endpoint was detected by achieving the initial value of the indicator current. Then, the timer was stopped, and the generating circuit was turned off. The number of electrons participating in the reaction between the acylhydrazones and the electrogenerated bromine was calculated using Faraday’s formula. The total antioxidant capacity was expressed as the quantity of electricity required for the titration of 1 mole of the compound.

#### 3.2.8. Statistical Analysis

The data were expressed as means ± SEM. Statistical comparisons were made using one-way analysis of variance (ANOVA), followed by Dunnett’s multiple comparison tests. Two-way repeated measures (mixed model) ANOVA followed by Bonferroni posttests were also used to compare recognition accuracy for the two objects. A difference with a *p*-value ≤ 0.05 was considered statistically significant. The statistical analysis was performed using GraphPad Prism 5 (GraphPad Software, San Diego, CA, USA).

Statistical analysis of the data was performed using Past 4.17 and Microsoft Office Excel 2016. The significance of differences in the mean values was determined using the Mann–Whitney U test (*p* ≤ 0.05), taking the Bonferroni correction into account. The IC_50_ and HC_50_ values were calculated using the MLA—Quest Graph™ IC50 Calculator AAT online calculator, C/O B&C Associates Limited Concorde House, Grenville Place, Mill Hill, London (Inc., 13 March 2025).

## 4. Conclusions

A series of water-soluble acylhydrazones was synthesized by reacting analogs of Girard’s reagents with bulk catecholaldehydes. The new compounds exhibited high antimicrobial activity against the Gram-positive bacteria *S. aureus*, *B. cereus*, and methicillin-resistant *S. aureus* strains. The leading compounds—isopropyl derivatives **9a** and **11a** and *tert*-pentyl analogue **10b**—exhibited activity against *S. aureus*, *B. cereus*, and methicillin-resistant *S. aureus* strains at a level of three times or greater than that of norfloxacin. The results of the CV assay showed that the most active compounds can only destroy the cell wall of *S. aureus* in high concentrations, indicating a different mechanism of antimicrobial action.

The compounds analyzed exhibited high bactericidal activity against the phytopathogens *M. luteus*, *P. atrosepticum*, *P. carotovorum* subsp. *carotovorum*, *Ps. fluorescens*, and *X. campestris*. Hydrazones **9a**,**b**,**d**; **10a**,**b**,**d**; and **11a**,**b**,**d** showed highly potent antiphytopathogenic attributes, with MBC values superior to those of the standard antibiotics. Compounds **9a** and **11a** can be said to possess outstanding bactericidal activity characterized by an inhibition zone width of 14 to 18 mm in the well-agar diffusion assay.

All the compounds showed high antioxidant activity, exceeding that of the industrial antioxidant BHT by 1.3–1.8 times, as well as that of pyrocatechol, indicating a significant contribution of the NH proton to the antioxidant activity of the synthesized compounds. All the compounds exhibited low cytotoxic activity and had no negative effects on the hemostatic system. This is evidence of their safety in relation to biological objects, positioning them as promising candidates in the search for new antimicrobial agents.

The compounds studied showed different effects on the hemostasis system. Compound **10b** exhibited antiplatelet activity that was 1.6 times greater than that of acetylsalicylic acid. Several of the compounds showed antiplatelet activity matching the level of acetylsalicylic acid. However, compound **9d** exhibited a pro-aggregant effect, increasing platelet aggregation by 6.3% and reducing the latent period by more than 15% compared to the control. Thus, our results can serve as the basis for future pharmaceutical efforts to create effective drugs with broad antithrombotic and hemostatic potential.

## Data Availability

The data are contained within this article.

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
