# Peer review of "Ammonium Catecholaldehydes as Multifunctional Bioactive Agents: Evaluating Antimicrobial, Antioxidant, and Antiplatelet Activity"

_ijms, 2025, doi:10.3390/ijms26167866_

Round 1
Reviewer 1 Report
Comments and Suggestions for Authors
Reviewer Report
This manuscript presents the synthesis and comprehensive biological evaluation of a series
of water-soluble ammonium catecholaldehyde-derived acylhydrazones. The compounds
were thoroughly characterized by NMR, IR, and elemental analysis. Their antimicrobial,
antioxidant, antiplatelet, and phytopathogenic activities were evaluated, showing
promising multifunctional properties. The work is generally well-structured, and the
experimental section is extensive. The study is potentially relevant for the development of
new multifunctional therapeutic agents.
Comments:
- The manuscript contains numerous grammatical errors and long sentences that impede
readability. For example, the abstract and introduction would benefit from professional
editing to improve clarity and flow. - Some figure captions are too brief. For instance, Figure 4 lacks an explicit explanation of abbreviations and symbols.
- In Table 2, clarify the notation “n.d.” (presumably “not determined”).
- The numbering of schemes and figures sometimes appears inconsistent between the
main text and supplementary materials—please double-check cross-references. - The authors may consider including SMILES strings or graphical depictions of the key
compounds in the main text to assist readers in visualizing structures. - Although the introduction provides references supporting hydrazone and catechol
pharmacophores, the rationale for the specific substituents and choice of ammonium
groups could be elaborated further. For example, why were trialkylammonium, DABCO-
based, and pyridinium moieties prioritized? Was this guided by prior SAR studies or
computational modeling? - While the manuscript provides mean MIC/MBC values and standard deviations,
statistical significance between compounds and controls is not consistently reported,
particularly in Table 4 and Table 5. Please, consider adding p-values or confidence
intervals to support claims of superior activity. - Figures illustrating MIC and MBC (Figure 9) are dense and difficult to read. The authors
should consider presenting MIC and MBC separately, with clearer labels and error bars. - The manuscript reports that compound 10b exceeded acetylsalicylic acid’s
antiaggregation activity. However, the biological relevance of the observed effects should be contextualized further. For example, how do the tested concentrations
compare to therapeutically relevant doses? Are there cytotoxicity risks at those
concentrations? - Also, compound 9d demonstrated proaggregant effects, which should be more explicitly discussed as a potential safety concern if such compounds were to be advanced toward therapeutic use.
- While the CV assay provides insight into membrane-disruptive activity, the mechanism
of antimicrobial action remains speculative. The authors should clarify whether further
studies (e.g., TEM imaging or molecular modeling) are planned to elucidate the mode
of action.
Reviewer Report
This manuscript presents the synthesis and comprehensive biological evaluation of a series
of water-soluble ammonium catecholaldehyde-derived acylhydrazones. The compounds
were thoroughly characterized by NMR, IR, and elemental analysis. Their antimicrobial,
antioxidant, antiplatelet, and phytopathogenic activities were evaluated, showing
promising multifunctional properties. The work is generally well-structured, and the
experimental section is extensive. The study is potentially relevant for the development of
new multifunctional therapeutic agents.
Comments:
- The manuscript contains numerous grammatical errors and long sentences that impede
readability. For example, the abstract and introduction would benefit from professional
editing to improve clarity and flow. - Some figure captions are too brief. For instance, Figure 4 lacks an explicit explanation of abbreviations and symbols.
- In Table 2, clarify the notation “n.d.” (presumably “not determined”).
- The numbering of schemes and figures sometimes appears inconsistent between the
main text and supplementary materials—please double-check cross-references. - The authors may consider including SMILES strings or graphical depictions of the key
compounds in the main text to assist readers in visualizing structures. - Although the introduction provides references supporting hydrazone and catechol
pharmacophores, the rationale for the specific substituents and choice of ammonium
groups could be elaborated further. For example, why were trialkylammonium, DABCO-
based, and pyridinium moieties prioritized? Was this guided by prior SAR studies or
computational modeling? - While the manuscript provides mean MIC/MBC values and standard deviations,
statistical significance between compounds and controls is not consistently reported,
particularly in Table 4 and Table 5. Please, consider adding p-values or confidence
intervals to support claims of superior activity. - Figures illustrating MIC and MBC (Figure 9) are dense and difficult to read. The authors
should consider presenting MIC and MBC separately, with clearer labels and error bars. - The manuscript reports that compound 10b exceeded acetylsalicylic acid’s
antiaggregation activity. However, the biological relevance of the observed effects should be contextualized further. For example, how do the tested concentrations
compare to therapeutically relevant doses? Are there cytotoxicity risks at those
concentrations? - Also, compound 9d demonstrated proaggregant effects, which should be more explicitly discussed as a potential safety concern if such compounds were to be advanced toward therapeutic use.
- While the CV assay provides insight into membrane-disruptive activity, the mechanism
of antimicrobial action remains speculative. The authors should clarify whether further
studies (e.g., TEM imaging or molecular modeling) are planned to elucidate the mode
of action.
Author Response
Q1. The manuscript contains numerous grammatical errors and long sentences that impede readability. For example, the abstract and introduction would benefit from professional editing to improve clarity and flow.
A1. Thank you! The request was taken into account. All sections of the manuscript were corrected.
Q2. Some figure captions are too brief. For instance, Figure 4 lacks an explicit explanation of abbreviations and symbols.
A2. Figure 4 is a general scheme of the work. Moreover, it does not contain any abbreviations. According to the recommendation, the titles of some figures have been detailed.
Q3. In Table 2, clarify the notation “n.d.” (presumably “not determined”).
A3. The footnote below table 2 already contains an explanation “n.d. - not determined”.
Q4. The numbering of schemes and figures sometimes appears inconsistent between the main text and supplementary materials - please double-check cross-references.
A4. Thanks for the reminder. The numbers of the figures in the text and supplementary do not match, as they are different. Each file uses its own numbering.
Q5. The authors may consider including SMILES strings or graphical depictions of the key compounds in the main text to assist readers in visualizing structures.
A5. The request was taken into account. SMILES strings were added to Table 1 for each compound obtained.
Q6. Although the introduction provides references supporting hydrazone and catechol pharmacophores, the rationale for the specific substituents and choice of ammonium groups could be elaborated further. For example, why were trialkylammonium, DABCO-based, and pyridinium moieties prioritized? Was this guided by prior SAR studies or computational modeling?
A6. Explanations of these issues are partly given in the Introduction (refs 16, 41, last paragraph) and Results and Discussion parts. In addition, it seemed appropriate to screen the bioactivity of target structures depending on the structure of the cationic center. In this work, we synthesized and studied hydrazones containing a structurally flexible (diethylmethyl), rigid bulky alicyclic (DABCO), and rigid planar (pyridinium) ammonium fragment.
Q7. While the manuscript provides mean MIC/MBC values and standard deviations, statistical significance between compounds and controls is not consistently reported, particularly in Table 4 and Table 5. Please, consider adding p-values or confidence intervals to support claims of superior activity.
A7. We have provided p-values for Table 4 and Table 5 to support claims of bactericidal and bacteriostatic activity values, respectively.
Q8. Figures illustrating MIC and MBC (Figure 9) are dense and difficult to read. The authors should consider presenting MIC and MBC separately, with clearer labels and error bars.
A8. It is Figure 8 that illustrates both MIC and MBC, the Reviewer points out to. In the revised version of manuscript, we have introduced a new paragraph just below the Figure 8 to underline that this scheme is for a general idea of both MIC and MBC variables at once measured for the all the compounds in five bacterial test-systems under question. We have also denoted that the error bars are omitted in the Figure 8 for more readable material, and then directly have gone to the Table 5 with MIC quantities supplied with the statistical significance characters between compounds 9-11 and reference antibiotics, in accordance to the Reviewer comments.
Q9. The manuscript reports that compound 10b exceeded acetylsalicylic acid’s antiaggregation activity. However, the biological relevance of the observed effects should be contextualized further. For example, how do the tested concentrations compare to therapeutically relevant doses? Are there cytotoxicity risks at those concentrations?
A9. The concentration adopted as a screening concentration corresponds to the therapeutic concentration and half the toxic concentration when taking acetylsalicylic acid orally at a dose of 75 mg. Added a link to the article. Thank you!
Q10. Also, compound 9d demonstrated proaggregant effects, which should be more explicitly discussed as a potential safety concern if such compounds were to be advanced toward therapeutic use.
A10. We have intentionally presented this compound to demonstrate that not only antiplatelet agents but also proplatelet agents can be found among the derivatives of this class of compounds, and this structure can also carry different mechanisms of action on the hemostasis system, which seems to us to be the most promising in isatin. Of course, this compound cannot be considered as an antiplatelet agent in the future, however, this result is important for other researchers who are developing antitumor or antimicrobial drugs, for example, which will be used in conditions of an altered hemostasis system. Moreover, hemostatic potential will be very useful wound heling.
Q11. While the CV assay provides insight into membrane-disruptive activity, the mechanism of antimicrobial action remains speculative. The authors should clarify whether further studies (e.g., TEM imaging or molecular modeling) are planned to elucidate the mode of action.
A11. Thank you for your careful consideration of our work and valuable comments. Indeed, the mechanism of antimicrobial action requires further study, and we fully agree with your proposal. As part of further research, we plan to conduct TEM visualization to study the ultrastructural changes in bacterial cells under the influence of the studied compounds, as well as molecular modeling to analyze their interaction with membrane components. These experiments will clarify the details of the mechanism of action and will be reflected in our future publications.

Reviewer 2 Report
Comments and Suggestions for Authors
Andrei V. Bogdanov and co-workers designed and synthesized a series of water-soluble acylhydrazones by reacting Girard's reagents with bulk catecholaldehydes. These compounds exhibited noticeable antimicrobial activity against gram-positive bacteria S. aureus, B. cereus and methicillin-resistant S. aureus strains.
Some of these compounds from the same series exhibited antibacterial activity, antioxidant and anti platelet activities. even though the activities are moderate to good, the article is missing the novelty. Compound 3 is known and reported in the literature and the designed compounds are analogs of compound 3.
In my opinion, the article doest meet the standards of the journal as it is missing the novelty aspect.
Below are the some the major comments to be considered.
1) The English used in the manuscript is poor and it need to be significantly improved (check for the grammar)
2) 1st paragraph in page 5 is so confusing (syn and anti differentiation is not clear) and the H1 and C NMRs explanation is so confusing.
Comments on the Quality of English LanguageThe English used in the manuscript is poor and it has to be significantly improved (check for the grammar)
Author Response
Andrei V. Bogdanov and co-workers designed and synthesized a series of water-soluble acylhydrazones by reacting Girard's reagents with bulk catecholaldehydes. These compounds exhibited noticeable antimicrobial activity against gram-positive bacteria S. aureus, B. cereus and methicillin-resistant S. aureus strains.
Some of these compounds from the same series exhibited antibacterial activity, antioxidant and antiplatelet activities. even though the activities are moderate to good, the article is missing the novelty. Compound 3 is known and reported in the literature and the designed compounds are analogs of compound 3.
In my opinion, the article doesn’t meet the standards of the journal as it is missing the novelty aspect.
A0: The aim of the work is to study the influence of the degree of spatial shielding of the hydroxyl groups of the pyrocatechol ring on the biological activity of compound 3 analogs, the high activity of which was established in our previous works [16, 41]. Indeed, this is part of our extensive research on the development of new agents with multitarget action.
We absolutely agree with the opinion that isatin is of natural origin and, at the same time, is a commercially and synthetically easily accessible compound. The ease of modification of the carbonyl group, aromatic fragment and nitrogen atom allows researchers in this field to consider this structure as a basic platform for medicine, organic synthesis, and chemistry of functional materials. From the point of view of medicinal chemistry, isatin is considered an object of “molecular hybridization” with the ability to control the properties of its derivatives as a result of modification of one or another reaction center. It is known that numerous derivatives of isatin exhibit various types of biological activity, such as anticancer, antituberculosis, antimicrobial, fungicidal, etc. However, in addition to our work, it should be noted that there is no data in the world literature on the study of the effect of a number of isatin derivatives on the hemostasis system, which seems to us to be a major drawback. Taking into account the above, we can conclude that the search for new antiplatelet and anticoagulant drugs based on functionalized isatins is relevant. Thus, all the obtained results can become the basis for future pharmaceutical developments to create effective drugs with broad antithrombotic and hemostatic potential and this result is important for other researchers who are developing antitumor or antimicrobial drugs, for example, which will be used in conditions of an altered hemostasis system.
Q1. The English used in the manuscript is poor and it need to be significantly improved (check for the grammar).
A1. The proposal was taken into account.
Q2. 1st paragraph in page 5 is so confusing (syn and anti differentiation is not clear) and the H1 and C NMRs explanation is so confusing.
A2. The proposal was taken into account. Some minor changes were made. For a clearer understanding and perception of the discussed isomerism, Figure 2 is given close to the text.
Reviewer 3 Report
Comments and Suggestions for Authors
Thus, all the obtained results can become the basis for future pharmaceutical developments to create effective drugs with broad antithrombotic and hemostatic potential and morever hemostatic potential will be very useful waound heling.
Author Response
Reviewer 3
Thus, all the obtained results can become the basis for future pharmaceutical developments to create effective drugs with broad antithrombotic and hemostatic potential and moreover hemostatic potential will be very useful wound heling.
A1. Thank you!

Round 2
Reviewer 1 Report
Comments and Suggestions for Authors
The authors have thoroughly addressed all my previous comments and suggestions and have revised the manuscript accordingly. In my opinion, the manuscript is now suitable for publication in the International Journal of Molecular Sciences.
Author Response
The authors have thoroughly addressed all my previous comments and suggestions and have revised the manuscript accordingly. In my opinion, the manuscript is now suitable for publication in the International Journal of Molecular Sciences.
A: Thank you!